# Sensory Acceptability and Proximate Composition of 3-Blend Plant-Based Dairy Alternatives

**DOI:** 10.3390/foods10030482

**Published:** 2021-02-24

**Authors:** Ama Frempomaa Oduro, Firibu Kwesi Saalia, Maame Yaakwaah Blay Adjei

**Affiliations:** Department of Nutrition and Food Science, College of Basic and Applied Sciences, University of Ghana, Legon, Accra 00233, Ghana; ama.oduro@gmail.com (A.F.O.); FSaalia@ug.edu.gh (F.K.S.)

**Keywords:** plant-based dairy alternatives, consumer acceptance, innovation, sustainable foods

## Abstract

Limitations of plant-based dairy alternatives as sustainable foods are their relatively low protein content and low sensory appeal. In this study, we used a consumer-led product development approach to improve the sensory appeal of existing prototypes of 3-blend dairy alternatives produced from melon seeds, peanuts and coconut. We used Relative Preference Mapping (RPM) and consumer acceptance testing using the 9-point hedonic scale to respectively identify innovative flavours and deduce the effect of ingredient components on consumer sensory appeal. Mixture design was used as the formulation tool to obtain optimized prototypes of the 3-blend dairy alternatives. Proximate analysis of the new prototypes, instrumental color assessment and consumer testing provided a basis to select a sustainable 3-blend dairy alternative. This prototype had a relatively high protein content (2.16%), was considered innovative by target consumers and also had a moderate liking score (6.55 ± 1.88) on the 9-point hedonic scale. Prototypes with higher protein content had low sensory appeal and were not considered innovative. Other prototypes with innovative sensory appeal had low protein content. By combining different plant raw materials and utilizing different sensory testing methods, we were able to design sustainable plant-based dairy alternatives which can be further optimized.

## 1. Introduction

The Food and Agriculture Organization (FAO) defines a sustainable diet as a diet that is nutritionally adequate, affordable, safe, and culturally acceptable while sparing natural and human resources [1]. The growing world population calls for the need to find sustainable ways to meet the increasing demand for food, especially protein-based foods. A sustainable diet should take into consideration not only the environmental impact but the nutrient density and adequacy of the diet [2,3,4,5]. Using livestock and their products to feed the world has a high environmental impact which includes the emission of greenhouse gases (GHGs) with the resultant negative impact on climate. Additionally, there is stress on the global nitrogen cycle and a negative effect on biodiversity. Water pollution, acidification, eutrophication and other adverse environmental impacts also result from rearing livestock [6,7,8,9,10,11]. Furthermore, the conversion of plant protein sources to animal protein for human consumption is only about 15% efficient [6,12,13] and this is not an environmentally sustainable process. There is thus a pressing need to find sustainable ways to feed the world with nutrient-rich diets that do not rely heavily on animal-sourced foods.

A plant-based diet is recommended as a means of mitigating the effects of relying on animal and animal products to meet the protein and nutritional needs of a growing world population [7,14]. Consuming a plant-based diet has a less negative environmental impact, uses less land and water, is relatively less expensive and more abundant than animal-based diets. Though some researchers have advocated that a balanced plant-based diet can provide all the nutrients needed for everyday life [8,12], others believe that plant-based diets are nutritionally inferior to diets including animal products [15,16].

Dairy alternatives are also called by various names including plant milk, plant milk alternatives, plant-based milk alternatives or other similar variations, but in the European Union (EU) and Canada, the term ‘milk’ is only used for the “normal mammary secretion obtained from one or more milkings without either addition thereto or extraction therefrom” [17,18]. In this article, we will refer to the dairy alternatives as plant-based dairy alternatives. Plant-based dairy alternatives are water extracts of dissolved and/or disintegrated plant material which look like dairy milk [15,16]. They have become popular and are typically consumed by people who have dairy milk allergy, are lactose intolerant, who have a preference for the vegan diet or who want to consume lower calories as part of special diets [10,16]. They are good dairy alternatives in places where dairy milk is too expensive or scarce [19]. Most plant-based dairy alternatives have health-promoting ingredients such as dietary fiber, antioxidants, minerals, vitamins, flavonoids, etc. [16]. Analysis of various plant-based dairy alternatives has shown that, apart from soymilk, most have low or no protein (<0.5) [10,20]. Additionally, most plant-based dairy alternatives do not have appealing sensory profiles [10,15]. Blending of different plant materials to produce plant-based dairy alternatives may be one way to improve both the nutritional and sensory profile of these products [16,21].

In our previous study, one, two and three blend plant-based dairy alternatives using melon seeds, peanut and coconut were formulated based on the functionality and perceived health benefits of these plant materials, their low cost and wide availability in Ghana [22]. An opportunity to improve the acceptability of these prototypes using a consumer-led approach was identified and formed the basis of this study. Various methods have been applied to products and process optimization using consumer appeal. Traditionally, consumer acceptance testing is used at the end of the development phase to evaluate the acceptability of products. Although this method allows direct measurement of consumer appeal and allows selection of the most appealing product, the unidimensional approach of acceptance testing does not adequately identify what drives appeal and how to improve or position the winning product. Recently, the Relative Preference Mapping (RPM) method, developed for wine public tastings to highlight innovative products, has been identified as a useful tool to guide product development optimization processes [23]. The method allows for quick identification of innovative flavours within a product prototype when compared to a known reference product. This comparison allows rapid selection of improved flavour prototypes which may be launched or taken into further development. Although RPM is useful to identify innovative flavours, it is limited in detailing the drivers of liking, since no sensory verbalization step is included in the technique. Formulation designs (or mixture designs), in combination with consumer acceptance tests, can be used to specify the ingredients or processes, with their interactions, that drive consumer acceptability.

This work follows up a previous study in order to develop, optimize and characterize 3-blend plant-based dairy alternatives formulated from locally available, inexpensive ingredients that are abundant, using a consumer-led product development approach [22]. We applied RPM to identify the innovations in flavour developed from a combination of melon seeds, peanuts, tiger nuts and coconut milk, then used mixture regression analysis of the consumer acceptability scores generated from samples obtained from the mixture design, to understand how the ingredients or their interactions drive consumer appeal for the product prototypes.

## 2. Materials and Methods

### 2.1. Materials

Melon seeds (*Citrullus lanatus* L.), peanuts (*Arachis hypogea* L.), coconuts (*Cocos nucifera* L.) and tiger nuts (*Cyperus esculentus* L.) were purchased from local markets in Ghana. Xanthan gum (Micrite Group Gh Ltd, Accra, Ghana) buffers: KH_2_PO_4_ and K_2_HPO_4_ (SureChem Prototypes Ltd., Suffolk, UK) were obtained in Accra while Bromelain tablets (Source Naturals, Scotts Valley, CA, USA) were obtained from the United States of America. Even ultra-high temperature pasteurized (UHT) full cream milk, a commercial pasteurized dairy milk was purchased and sweetened to a specified level and used as the reference sample for the RPM consumer test. Vitamilk, a commercial soymilk beverage (a popular plant-based dairy milk alternative) was also purchased and included in the product test set.

### 2.2. Methods

#### 2.2.1. Product Formulation

Focus group discussions with target consumers (unpublished data) were used to glean information on product characteristics for an appealing plant-based dairy alternative. Three focus group discussions with seventeen (17) participants were organized. Participants were aged between 18 and 49 years and the three groups were made up of eight (8) females aged between 18 and 30 years, four (4) males aged from 18 to 30 years and a mixed group of five (5) males and females aged between 31 and 49 years. These took place in the meeting room of the sensory evaluation laboratory of the University of Ghana. Each discussion lasted for 90 min. The discussions centered around which plant materials to include and the expected sensory characteristics desirable in a plant-based dairy alternative for consumers. We incorporated product tasting during the discussions to guide consumer responses. Based on consumer feedback from the focus group discussions, the prototypes developed earlier by Odoom (2018) [22] were optimized in three key ways; ingredient change, process change and re-formulation using a four-component lattice mixture design from Minitab v. 17.1.0 (Minitab Inc., State College, PA, USA). The three elements are outlined as follows:Ingredient change: tiger nut was added as a new ingredient to the original ingredients (melon seeds, peanuts and coconut) to use as raw material.Process change: a new process was developed to process peanut milk; peanuts were roasted instead of soaking in 2% NaHCO_3_ for 24 h.Reformulation: a four-component mixture design using Minitab v. 17.1.0 (Minitab Inc., State College, PA, USA), instead of a three-component mixture design used in the previous study by Odoom (2018) [22].

The software (Minitab v. 17.1.0) generated 19 product formulations for the four-component mixture design. The upper and lower bound constraints for the four-component mixture design are shown in Table 1. The resulting 19 trial runs included 1-, 2-, 3- and 4-blend formulations as shown in Table 2. Only 3-blend prototypes were used for the consumer test using RPM, as we wanted to leverage on the nutritional and sensory characteristics of three plant-based dairy alternatives instead of 1 or 2 to produce a nutritionally adequate and consumer acceptable plant-based dairy alternative. A four blend plant-based dairy alternative was not investigated in this study.

#### 2.2.2. Processing of Plant-Based Dairy Alternatives

Each plant-based dairy alternative was extracted separately to obtain a single dairy alternative before blending them according to the predetermined ratios obtained from the mixture design (Table 2). The plant-based dairy alternatives were processed based on the method described by Odoom (2018) [22] with the following modifications: Bromelain enzyme was added to hydrolyze the proteins in the melon seed milk to control viscosity before blending in the formulations. This was done because Odoom (2018) [22] realized that melon seed milk coagulated when heated; this is caused by the unfolding of proteins and the exposure of non-polar amino acids to water, increasing the surface hydrophobicity. The increased protein to protein interaction leads to aggregation or gelling [10]. Another modification was that the peanuts were roasted at 120 °C for 40 min instead of soaking in 2% NaHCO_3_ for 24 h during peanut milk processing. This was done in response to consumer feedback from the focus group discussions. To produce tiger nut milk, tiger nuts were manually sorted to remove contaminated and defective nuts. The nuts were washed with water and roasted for 15 min at 120 °C in an electric convection fan oven (HC 62062, Kaiser, Berlin, Germany). The tiger nuts were blended with water in the ratio of 1:4 in an automatic home soymilk mixer (PB103, AliExpress Ana zhang Store, China) for 2 min. The slurry was filtered using a plastic kitchen mesh strainer to obtain tiger nut milk. To each single plant-based dairy alternative, weighed quantities of buffer, cane sugar and stabilizer were added, heated to 70 °C and homogenized by passing through a colloid mill (Premier 84, Premier Colloid Mills Limited, Walton on Thames, Surrey, UK).

To produce the various formulations, the four pretreated milk samples were mixed according to the ratios obtained using the four-component lattice mixture design (Table 2). The samples were pasteurized at 80 °C for 20 min. The pasteurized milk was hot-filled into 500 mL chlorine sterilized PET plastic bottles with caps and closed. They were rapidly chilled to 5 °C and stored at 4 °C.

#### 2.2.3. Identifying Products with Innovative Flavour Using RPM

Relative Preference Mapping was designed to be used in a social setting [23]; as such, this test was carried out at selected product and food fairs to depict a social setting as described by the developers of the method. The T-Map scale shown in Figure 1 was used for the evaluation of the prototypes. It has two axes, the *y*-axis which is the liking axis, and the *x*-axis which is the difference axis. The same test protocol as described for RPM using the T-Map scale was used [23]. The T-Map scale was printed on an A0 sheet, which was set up using a flip chart stand. Assessors were given a strip of colored dots with the different product codes: they were instructed to place the product codes on the T-Map scale based on their assessment. A total of 90 consumers of plant-based dairy alternatives participated in this study. Participants were selected by a one-on-one interview using a recruitment questionnaire. They were verbally screened for any known allergy to any of the ingredients used for the formulations. Participants chosen were those who indicated that they were consumers of plant-based dairy alternatives. They proceeded with the tasting after they had signed an informed consent form. Of the 19 runs obtained from Minitab, six (6) were 3-blend formulations and were the main test products in this study (Table 3). The RPM protocol requires a reference product which is repeated in the product set as a blind control product. In this study, Even ultra-high temperature pasteurized (UHT) full cream milk, a commercially available dairy milk product, was used as the reference, R, and blind control, BC. This product was sweetened to the same level of added sugar as the prototypes to reduce bias caused by sweetness as a source of consumer appeal. This was done by adding 4% *w/v* of granulated sugar to the dairy milk sample and stirring to ensure that all the sugar was dissolved. This was the same quantity of sugar added to the prototypes during processing. A limitation of this consumer test was that the extra sugar added to the dairy milk sample could have influenced consumer perception of the reference sample. Vitamilk, a popular and widely consumed plant-based dairy alternative, made from soybeans, was also added to the product set as sample V. This product is already sweetened and was used “as is”. Altogether, assessors evaluated eight (8) products in this test. Each assessor was served with 20 mL of the reference sample to familiarize themselves, after which 10 mL of each test prototype was served to them in a predetermined randomized order based on the William’s Latin Squares design in Compusense. Each assessor tasted the product set only once. They were allowed to re-taste the reference as often they wished during the evaluations and could be served more if required. All samples were served chilled at 10 ± 2 °C. They were maintained at this constant temperature by placing them in a chill box when they were not being tasted. To make a judgement, assessors were instructed to simultaneously evaluate how different was, and how much they liked, the test product compared to the reference product, after which they placed the corresponding product code on the T-map scale.

#### 2.2.4. Consumer Acceptance Test and Mixture Design Analysis to Understand Ingredient Contribution to Product Acceptance and to Identify Optimal Formulation

To explore which ingredients influence consumer appeal, mixture design regression analysis was used. Consumer acceptance scores served as the dependent value in the model and required that the full set of 19 prototype runs were evaluated by consumers. Considering the large number of products to be tasted by consumers, a Balanced Incomplete Block Design (BIBD) [24] method was used for this test. The BIBD design was set up using SAS^®^ v. 9.4 (100 SAS Campus Drive, Cary, NC, USA) and required 19 regular consumers of plant-based dairy alternatives to complete the taste test. Each assessor evaluated 9 out of the 19 prototypes. Assessors used the traditional 9-point hedonic scale with descriptors and numbers to score how much they liked each of the products. On this scale, a score of 1 means: dislike extremely and a score of 9 means like extremely. This was a paper-based test.

Each assessor received 15 mL of each prototype which was served in 20 mL disposable transparent plastic cups at a temperature between 8–10 °C. In this study, only the 19 prototypes were assessed; the commercial products (dairy milk sample (BC) and the plant-based dairy alternative, Vitamilk (V)) used in the RPM test were not included as the objective of this consumer test was not to compare the acceptability of the prototypes with these samples.

Assessors were asked to evaluate the prototypes based first on their overall liking of the prototype, after which they proceeded to evaluate how much they liked the appearance, flavour, mouthfeel, consistency and after taste sensations. After evaluating each product, assessors were asked to write down what they liked or disliked about each prototype. They rinsed their mouths with water before proceeding to the next prototype. The consumers used in this study were not the same as those who participated in the RPM study.

#### 2.2.5. Proximate Composition and Instrumental Colour Evaluation

##### Proximate Composition

The proximate composition of the prototypes, the commercial dairy milk alternative (V) and the dairy milk product (BC) were assessed. Total moisture was determined as described by Pinelli et al. (2015) [25]. For each beverage 20 mL was dried in a hot air oven at 105 °C till constant mass. The nitrogen content (N) of the samples was determined using the Kjeldahl method as described by the Association of Official Analytical Chemists (AOAC) (2005) [26] (method 991.20). The total protein was calculated by multiplying N (the total nitrogen) with 6.25 as the conversion factor. The total solids content was calculated by subtracting the total moisture content from 100%. The total fat content was determined using the Rose-Gottlieb Method AOAC (2005) [26] (method 905.02). The total ash was determined by heating 2g of the sample in a furnace at 600 °C for 6 h as described by AOAC (2005) [26] (method 935.45). Total carbohydrate content was determined by difference as described by Pinelli et al. (2015) [25]: Crude Carbohydrates = 100% − (%protein + %moisture + %crude fat + % crude ash).

##### Color

The color of the products used for RPM was measured as described by Aidoo et al. (2010) [27]; measurements were based on the L*a*b* color system and were determined using a Minolta Chroma Meter (Data Processor DP-301, for Chroma Meter CR-300 series). The color difference (ΔE) was determined using the following formula: (ΔE = (L* − L) 2 + (a* − a) 2 + (b − *b) 2)1/2. The standard white tile to which the samples were compared had the following color indices: (L* = 97.95, a* = −0.12, b* = +1.64).

The determinations were carried out in triplicate; means and standard deviations are reported.

#### 2.2.6. Data Analyses

##### Identifying Products with Innovative Flavours using RPM

The scores from RPM were analysed as described by Blay et al. (2012) [28]. The difference and liking axes of the T-Map scale were each 50 cm long with intervals of 1 cm. The scores were derived by reading the mark made by each assessor from the end of each axis. For the difference axis (*x*-axis), the end R was taken as the starting point and, as the mark moved away from R, this was considered more different from the reference. For the liking axis, the starting point was at the bottom of the *y*-axis and increased as the data points moved up. Lower values meant that the product was disliked or was similar compared to the reference and high values meant that the product was liked and was different compared to the reference. Data from the liking axis and difference axes were analyzed separately using a one-way analysis of variance (ANOVA). Individual product maps created by consumers were condensed into a consensus product map using the Generalized Procrustes Analysis (GPA) method. The region of innovation was highlighted as the space between the liking and difference axes on the consensus map. All analyses of RPM data were carried out using XLSTAT (Addinsoft 40, rue Damrémont 75018, Paris, France).

##### Understanding the Role of Ingredients in Product Acceptability

Regression analysis for mixture design using Minitab v. 17.1.0 was performed. The dependent variable for the regression models were the consumer acceptance scores obtained from the BIBD study using the 9-point hedonic scale. The independent variables were coconut milk (X_1_), peanut milk (X_2_), tiger nuts milk(X_3_) and melon seed milk (X_4_). Cox response plots were generated for overall liking, appearance, flavour, mouthfeel, consistency and aftertaste. 

Product liking scores were analyzed using a one-way analysis of variance (ANOVA). Post-hoc analysis was based on Fisher’s Least Significant Difference (LSD).

## 3. Results and Discussion

### 3.1. Proximate Composition of the 3-Blend Prototypes

The proximate composition of the prototypes, the commercial plant-based dairy alternative and the dairy milk product are shown in Table 4. The products are arranged in descending order of protein content as this component is of interest to our study. Overall, the plant-based dairy alternatives had protein content of 1.65% to 3.51%, fat content of 4.71% to 6.54%, an ash content of 0.25% to 0.49 and carbohydrate content of 4.28% to 7.26%. The moisture content was between 85.26% to 88.22% and total solids raged between 11.78% and 14.74%. Both commercial products had moisture and total solids content within this range.

Not surprisingly, the animal-sourced milk product, Product BC, had the highest protein content of 3.76%. Within the 3-blend prototypes, only product H (M_37.5%_, C_25%_, P_37.5%_) had protein content comparable to the dairy milk product, (3.51%) which was higher than the protein content for the commercial soymilk product V (3.30%). This is an important observation as it supports the notion that blending different plant materials could increase their overall protein content. Melon seeds and peanuts have considerably high protein content given that they are legumes. Peanuts have a protein content of 23.68% [16] while melon seeds have a protein content of 25.4% [29]. As such those prototypes with high contents of melon seeds and peanut milk also had considerably high protein contents. Prototype H (M_37.5%_, C_25%_, P_37.5%_) had the highest combination of melon seed milk and peanut milk in its formulation at 37.5% each with only 25% of coconut milk. Coconut and tiger nuts have a relatively lower protein content, 2.60% [30] and 5.04% [31], respectively, but higher fat and carbohydrate content. As such prototypes with high components of these ingredients had lower protein content but higher carbohydrate or fat content. Other prototypes with relatively high protein contents included prototype R (M_37.5%_, C_25%_, T_37.5%_) at 2.31% and N (M_25%_, C_50%_, P_25%_) at 2.16%. Prototype R (M_37.5%_, C_25%_, T_37.5%_) has a high melon seed content of 37.5%. Although N (M_25%_, C_50%_, P_25 %_) has a high coconut milk content, its protein content was improved as a result of the blend of melon seeds and peanut milk. The prototypes with the lowest protein contents had high amounts of coconut milk and no or low melon seed milk or peanut milk. Specifically prototypes P (C_50%_, T_25%_, P_25%_), A (C_50%_, T_25%_, M_25%_) and E (C_25%_, P_37.5%_, T_37.5%_) had the lowest protein contents of 1.65%, 1.89% and 1.90, respectively.

Dairy milk contains all the essential amino acids in their right proportions [32]. Peanut contains all the essential amino acids except for methionine. Cysteine is also a limiting amino acid in peanuts. The quantities of the other essential amino acids are comparable to that of dairy milk [33,34]. Coconut has all the essential amino acids but their quantities are lower than those found in dairy milk [34], although their quantities meet the FAO/WHO requirements for adults [35]. Glutamic acid and arginine are the most abundant amino acids in coconut [35,36]. Melon seeds also contains all the essential amino acids but lysine is the limiting essential amino acid and cysteine is found in low quantities [37]. Tiger nuts contains all the essential amino acids. A study of the amino acid profile of tiger nut by Rasaq et al. (2013) [38] indicated that tiger nut milk meets the amino acid requirement for both adults and children. Even when plant raw materials contain all the essential amino acids, plant proteins are not as easily digestible as animal-based proteins [10]. Prototypes containing peanuts, tiger nuts and coconuts can have all the essential amino acids though not of the same quality and proportions found in dairy milk, while prototypes with melon seeds, peanuts and coconuts will still lack cysteine. A combination of tiger nuts, melon seed and coconut milk will also have all the essential amino acids.

Coconut and tiger nut milk contribute to a high fat content in the final formulation since they have high fat contents of (21.33%) [39] and 24.49% [31], respectively. The products with high quantities of peanut milk had the lowest ash contents, and this was also the trend when Aidoo et al. (2010) [21] combined peanuts and cowpea to replace skimmed milk powder in chocolate. Those with high amounts of melon seed milk without peanut milk had higher ash content. Dairy milk has a calcium content of between 122 mg–134 mg/100 g [34], tiger nuts have a calcium content of 40 mg/100 g [40], peanuts, coconuts, melon seeds have a calcium content of 54 to 92 mg/100 g [34] and 18.1 mg/100 g [41] and 28.2 mg/100 g [42], respectively. Because of these low calcium levels in most plant raw materials, most plant-based dairy alternatives have to be fortified with calcium [10]. Prototypes with high levels of tiger nuts milk had high carbohydrate contents as tiger nuts have high carbohydrate content (43.3%) which is mainly made up of starch [31]. The relevance of blending different plant materials to develop a sustainable dairy milk substitute is the improved protein content of the final formulation. Notwithstanding, a sustainable diet should, in addition to being environmentally friendly, be nutritionally adequate and have an acceptable sensory appeal for consumers.

### 3.2. Colour of the 3-Blend Prototypes

With regards to product appeal, appearance is the first attribute of choice, although repeat purchase is most influenced by the flavour and taste of the product. A visual inspection of the extracts from the four materials used to formulate the 3-blend prototypes show that they appear whitish, cream and reddish-brown (Figure 2). Instrumental color assessment, however, provides an objective measurement. With regards to the appearance of milk, the lighter the color, the more acceptable it tends to be for consumers; however, too much lightness might reduce consumer appeal. The Hunter Lab Chromameter measured the degree of lightness L* and the hue; red-green (a-value) and yellow-blue (b-value). The L* value measures the lightness of a product. The higher the L* value, the whiter the product. The b* value when positive signifies a high yellowness of the product and when negative connotes a blue color. High positive a* values indicate redness, whilst a low or negative a* value indicates greenness. Generally, all the prototypes were light and had L* values ranging between 75.91 and 85.54 (Table 5). The dairy milk sample (BC) seen in Figure 3 had the highest L* value of 95.23 compared to the prototypes. The a* values were low and negative (in the green spectrum). The b* values were low and positive (in the yellow spectrum).

There were statistically significant differences between all the measured color indices of the different prototypes. Prototypes that contained high quantities of coconut milk had higher L* values while those that contained high quantities of tiger nut milk had lower L* values. Product BC had the highest L* value of 95.23 and amongst the dairy milk alternatives, Prototype N (C_50%_, P_25%_, M_25%_), which had a high percentage of coconut milk and no tiger nut milk, had the highest L* value of 85.54. Prototype R (C_25%_, T_37.5%_, M_37.5%_) had the lowest L* value (75.91), while prototype E (C_25%_, P_37.5%_, T_37.5%_) had the second lowest L* value (78.22). Both prototypes contain high percentages of tiger nut milk and low amounts of coconut milk in their formulations which could account for the darkness of the prototypes. That none of the formulations had lightness value exceeding that of commercial dairy milk is a positive outcome, as they will not appear unnaturally white to detract from its consumer appeal.

### 3.3. Identifying Products with Innovative Flavours Using RPM

Data from RPM was analysed as described by Adjei et al. (2020) [23] by generating a consensus product map using GPA. The area of innovation is the area where products are loaded in the area of the angle formed between the difference and liking axes on the 2D map. In this study, when compared to the commercial animal sourced milk (BC), the products that were considered innovative flavours were products E (C_25%_, P_37.5%_, T_37.5%_), P (C_50%_, P_25%_, T_25%_), and V (Figure 4). These products had no melon seeds but had coconut and peanuts. The commercial product made from soybeans was also considered innovative in flavour. Prototype N (C_50%_, P_25%_, M_25%_) fell on the border of the area of innovation.

According to Gruneert et al. (1997) [43], an innovative product is that which is perceived by people as new. Johannessen et al. (2001) [44] asserted that innovation depends on the people assessing the product; as such, what could be considered an innovation by someone might not be considered so by another individual or organization. In our research, an innovative product was one which was different from the reference (commercial milk product, BC) and liked enough by consumers to load into the “area of innovation”. Although products V (Soymilk), prototypes E (C_25%_, P_37.5%_, T_37.5%_) and P (C_50%_, P_25%_, T_25%_) were the least different from the reference product, they were different enough to be considered innovations by consumers and their difference from reference scores were statistically different from the dairy milk product, BC (Figure 5). Diarra et al. (2005) [45] suggested that plant-based dairy alternatives that have similar sensory characteristics to dairy milk were more accepted, and Sakthi et al. (2020) [46] found that the more similar the sensory profile of a dairy milk alternative is to dairy milk, the more it is accepted. Prototypes E (C_25%_, P_37.5%_, T_37.5%_) and P (C_50%_, P_25%_, T_25%_) are comparable to the already commercial product V (Vitamilk), which also loaded in the innovation area, and with further optimization could be potential new products for the dairy alternative market. The least liked prototypes H (C_25%_, P_37.5%_, M_37.5%_) and R (C_25%_, T_37.5%_, M_37.5%_) were also perceived as the most different.

Figure 6 shows the liking scores of the experimental prototypes compared to the dairy milk reference (Even ultra-high temperature pasteurized (UHT) full cream milk). Product BC (Even ultra-high temperature pasteurized (UHT) full cream milk) and V (commercial dairy milk alternative) were the most liked, and there was no statistical difference between their liking scores. All the 3-blend prototypes were liked less than the two commercial products tested, suggesting that there is still room for improvement in this current formulation of the 3-blend products to obtain an optimum product. Prototypes E (C_25%_, P_37.5%_, T_37.5%_) and P (C_50%_, P_25%_, T_25%_) were the most liked amongst the 3-blend prototypes and their liking scores were not statistically significantly different from each other. Thus, either of these two products could be considered innovative flavours; however, these two products had the lowest protein content when the proximate analyses were done. Further improvement in this formulation will be required to improve its protein content while retaining its acceptance as an innovation in flavour. Product H (C_25%_, P_37.5%_, M_37.5%_) was the least liked prototype, although this was the product with the highest protein content when the proximate concentrations were evaluated. This product was not considered an innovation at all. However, prototype N (C_50%_, P_25%_, M_25%_), which fell on the borderline of the innovation area, could be highlighted as a potential product for optimization as a sustainable plant-based dairy alternative since it had a relatively higher protein content 2.16% and shows some innovation. Product R (C_25%_, T_37.5%_, M_37.5%_) had a higher protein content than N (C_50%_, P_25%_, M_25%_); however, it did not fall in the innovation area although there was no significant difference in its liking scores compared to Product N (C_50%_, P_25%_, M_25%_). It is possible that the color of product R (C_25%_, T_37.5%_, M_37.5%_), may have influenced its position on the consensus map as the instrumental color assessment shows a significant difference between product R (C_25%_, T_37.5%_, M_37.5%_) and N (C_50%_, P_25%_, M_25%_), with product R (C_25%_, T_37.5%_, M_37.5%_) having the lowest lightness intensity (Table 5). The advantage of the RPM method is that it allows evaluation of product innovation on a two-dimensional axis as opposed to the one-dimensional approach of using only the traditional liking scale. Thus, true innovations in product sets are evaluated on a holistic basis when a relevant reference product is selected.

### 3.4. Understanding the Role of the Ingredients That Drive Liking

#### 3.4.1. Effect of the Ingredients on Overall Liking Scores

Figure 7 shows the Cox response trace plot for overall liking. As the quantity of coconut and peanut milk is increased in the formulation, the overall liking scores rise; on the other hand, tiger nut milk increases the overall liking scores as its proportion increases up to a point (+0.30 deviation from the reference blend) and then decreases liking scores as its proportion continues to increase. Melon seed milk, however, leads to a reduction in the overall liking scores of the product as its proportion increases in the formulation.

Table 6 shows the regression coefficients for the different sensory properties. Although strong positive regression coefficients for overall liking and the other modalities were observed, they did not have statistical significance at the 95% confidence level (apart from consistency and after taste liking). This is probably due to the reduced number of replicates using the BIB design instead of a complete block design. The observations may be considered to be trends, as similar observations have been described elsewhere. For instance, in this study, coconut milk and peanut milk have the most positive effect on overall liking. This observation is similar to other findings using these ingredients, such as Obinna-Echem and Torporo (2018) [47] observed about coconut milk increasing acceptability scores for tiger nut/coconut milk blends. Likewise, Sakthi et al. (2020) [46] found that roasted peanut milk had the best sensory acceptability amongst different pre-treatments for peanut milk preparation when comparing soaking, germination, blanching and roasting. Melon seed milk has a less positive effect, though its effect was greater than tiger nut milk. The interactions between coconut and peanut milk, coconut and melon seed milk, and peanut and melon seed milk had a negative effect on the overall liking scores, while that between coconut and tiger nut milk, peanut and tiger nut milk and tiger nut and peanut milk had a positive effect on the overall liking scores.

Table 7 shows the liking scores for the 19 formulations. 100% coconut milk, prototype L(C_100%_) had the highest overall liking score of 7.89; prototypes containing high quantities of coconut milk and peanut milk had high overall liking scores, for example, prototypes E (C_25%_, P_37.5%_, T_37.5%_), G (C_25%_, P_75%_) and I (C_62.5%_, T_37.5%_) had overall liking scores of 7.44, 7.44 and 7.22, respectively. Even though tiger nut milk interacts positively with the other components, prototypes with tiger nut milk above 50% had low overall liking scores; an example of this is prototype F (C_25%_, T_75%_) which had an overall liking score of 4.44. Prototypes with coconut milk, peanuts and tiger nut milk had high overall liking scores. However, those with high quantities of melon seed milk had low liking scores. The prototypes with the lowest scores for overall liking had high melon seed milk. This is because melon seed milk interacts negatively with the other components. This is exemplified by prototypes O (C_62.5%_, M_37.5%_), D (C_34.4%_, P_9.4%_, T_9.4%_, M_46.9%_) and H (C_25%_, P_37.5%_, M_37.5%_), whose liking scores were 2.77, 4.0 and 4.11, respectively. Prototype J (C_43.8%_, P_46.9%_, T_9.4%_, M_9.4%_) had the lowest overall liking scores of 2.56, even though it had high quantities of peanut and coconut milk; the high quantities of both coconut and peanut milk might have led to a negative interaction as their regression coefficient is negative (−5.01), as seen in Table 6. The results of this consumer test agree with the results of the RPM as the 3−blend prototypes with coconut milk, peanut and tiger nut milk had high liking scores while those with melon seed milk had low overall liking scores. Only prototype P (C_50%_, P_25%_, T_25%_) had lower than expected overall liking scores.

#### 3.4.2. Effect of the Ingredients on Appearance Liking

Figure 8 shows the Cox response trace plot for appearance liking. Melon seed and tiger nut milk cause a decline in liking scores as their proportion increases. Tiger nut milk lowers the liking scores more; this could be due to its darker color compared to the other plant−based dairy alternatives (Figure 2). Coconut milk increases liking scores up to a point (+0.40 deviation from the reference blend), then reduces them as its amount increases; coconut milk had an extremely white color (Figure 2) that did not look like the natural color of milk, which had the potential to make the prototypes too light in color if the proportion deviates by about +0.40 from the reference blend.

The regression coefficients (Table 6) show coconut milk, peanut milk and melon seed milk have higher positive effects on the appearance liking scores compared to the lower positive effect that tiger nut milk has. The interactions between tiger nut and peanut milk, and tiger nut and coconut milk were positive as they made the resulting prototype lighter, but the interaction between peanut and melon seed milk, and tiger nut and melon seed milk were negative. The appearance liking scores (Table 7) show that M (C_62.5%_, P_37.5%_), C (C_71.9%_, P_9.4%_, T_9.4%_, M_9.4%_) and O (C_62.5%_, M_37.5%_) had the highest liking scores of 8.11, 7.78 and 7.67, respectively; these contain high quantities of coconut milk and low or no tiger nut milk. Prototype N (C_50%_, P_25%_, M_25%_) had the highest L* score (85.54) amongst the 3−blend prototypes and loaded on the borderline of the innovation area on the product map (Figure 4); in contrast, prototype R (C_25%_, T_37.5%_, M_37.5%_), which was liked as much as N (C_50%_, P_25%_, M_25%_) in the consumer test using RPM (Figure 5), was not innovative and was amongst the products which scored lowest for appearance liking (Table 7). This confirms the earlier assertion that the color of the product is an important factor which affects the appeal of the product. 

#### 3.4.3. Effect of the Ingredients on Flavour Liking

Figure 9 shows the effect of the ingredients on flavour liking scores. Melon seed milk decreases the liking scores as its proportion increases, but appears to increase these scores as its quantity deviates from the reference blend by +0.55, whilst the effect of tiger nut milk is neutral up to the 0.0 deviation from the reference formulation, then decreases the liking scores for flavour as its deviation from the reference formulation increases. On the other hand, as the amounts of coconut and peanut milk increase, the liking scores for flavour rise.

The regression coefficients (Table 6) show that peanut and coconut milk have the most positive effect on the flavour liking scores, respectively. Melon seed milk has a slightly higher positive effect on the flavour scores compared to tiger nut milk. The interactions between coconut and tiger nut milk, peanut and tiger nut milk and tiger nut and melon seed milk have a mild positive effect on flavour liking scores while those between coconut and peanut milk, coconut and melon seed milk, and peanut and melon seed milk have a higher negative effect on flavour liking scores. The liking scores for flavour show that prototypes L (C_100%_), G (C_25%_, P_75%_), and E (C_25%_, P_37.5%_, T_37.5%_) have the highest liking scores with scores of 7.78, 7.22 and 7.0, respectively; these contain mainly coconut and peanut milk with some tiger nut milk and without any melon seed milk; while the prototypes with the lowest scores for flavour D (C_34.4%_, P_9.4%_, T_9.4%_, M_46.9%_), J (C_43.8%_, P_46.9%_, T_9.4%_, M_9.4%_), and O (C_62.5%_, M_37.5%_), with liking scores of 3.78, 4.11 and 4.22, have high quantities of melon seed and tiger nut milk and show negative interaction between peanut milk and coconut milk.

#### 3.4.4. Effect of the Ingredients on Mouthfeel Liking

Figure 10 shows the effect of the ingredients on the mouthfeel liking scores of the prototypes. Melon seed milk decreases the liking scores as its volume increases but appears to improve these scores as its proportion increases up to a point (+0.55 deviation from the reference blend), whilst tiger nut milk shows the opposite effect; it increases the liking scores up to the 0.0 deviation from the reference blend, then decreases liking scores as its amount increases in the blend. Coconut milk and peanut milk both increase the liking scores as their quantity increases.

The regression coefficients (Table 6) show that coconut milk, peanut milk and melon seed milk have higher positive effects on the mouth feel liking scores compared to the lower positive effect that tiger nut milk has on the liking scores. The interactions between coconut and tiger nut milk, peanut and tiger nut milk and tiger nut and melon seed milk are positive, while those between coconut and peanut milk, coconut and melon seed milk and peanut and melon seed milk are negative. Table 7 shows that the prototypes with the highest scores for mouth feel were L (C_100%_), E (C_25%_, P_37.5%_, T_37.5%_), and G (C_25%_, P_75%_) with scores of 7.78, 7.44 and 7.44, respectively, while those with the lowest scores were D (C_34.4%_, P_9.4%_, T_9.4%_, M_46.9%_), J (C_43.8%_, P_46.9%_, T_9.4%_, M_9.4%_), and O (C_62.5%_, M_37.5%_), with scores of 4.11, 4.33 and 4.44, respectively. This shows that those prototypes that contain high quantities of either coconut or peanut milk had high scores for mouthfeel, while those with high volumes of melon seed milk had low mouthfeel liking scores.

#### 3.4.5. Effect of the Ingredients on Consistency Liking

Figure 11 shows the effect of the ingredients on consistency liking scores of the prototypes. Melon seed milk decreases the liking scores as its proportion increases, but appears to cause these scores to rise as its amount increases at +0.3 deviation from the reference blend, while the effect of tiger nut milk is neutral to a point (0.0 deviation from reference blend), then causes the liking scores to plunge as its proportion increases. Coconut milk and peanut milk both increase the liking scores as their quantity increases.

The regression coefficients (Table 6) show that peanut milk and coconut milk have the most positive effect on the consistency liking scores, respectively. Melon seed milk has a higher positive effect on the consistency liking scores compared to tiger nut milk. The interaction between coconut and tiger nut milk was the only positive interaction between the components. The negative interactions between coconut and melon seed milk (−6.567) and peanut and melon seed milk (−6.724) were significant at the 95% confidence level. The prototypes with the highest scores for consistency G (C_25%_, P_75%_), L (C_100%_), and M (C_62.5%_, P_37.5%_), 7.44, 7.33 and 7.0, respectively, contained no melon seed milk. Those with the lowest scores F (C_25%_, T_75%_), R (C_25%_, T_37.5%_, M_37.5%_) and S (C_25%_, P_25%_, T_25%_, M_25%_), with scores of 4.89, 5.0. and 5.22, respectively, contained melon seed milk and a high amount of tiger nut milk.

#### 3.4.6. Effect of the Ingredients on Aftertaste Liking

Figure 12 shows the effect of the ingredients on the aftertaste liking scores of the prototypes. Melon seed milk decreases liking scores for aftertaste as its proportion increases; while tiger nut milk increases the liking scores to a point (0.35 deviation from reference blend) and starts decreasing these scores as its proportion increases. Peanut milk and coconut milk on the other hand increase the liking scores.

Table 6 shows the regression coefficients for aftertaste. These show that peanut and coconut milk have the most positive effect on the consistency liking scores, respectively. Melon seed milk has a higher positive effect on the aftertaste scores compared to tiger nut milk. The interaction between coconut and melon seed milk has a significantly negative (−7.945) effect on the aftertaste of the prototypes. Prototypes L (C_100%_), G (C_25%_, P_75%_) and E (C_25%_, P_37.5%_, T_37.5%_) had the highest scores for aftertaste, at 7.78, 7.11 and 7.0, respectively, while J (C_43.8%_, P_46.9%_, T_9.4%_, M_9.4%_), O (C_62.5%_, M_37.5%_) and H (C_25%_, P_37.5%_, M_37.5%_) had high quantities of melon seed milk and had low aftertaste liking scores of 3.0, 3.67 and 3.78, respectively.

A limitation of this study was the low power of the consumer test using BIB, as each prototype was tasted only nine times.

## 4. Conclusions

Proximate analysis showed that products that were high in melon seed and peanut milk had higher protein contents, but those products had low consumer acceptance, while the products without melon seed were considered innovative but had a low protein content. Only prototype N (C_50%_, P_25%_, M_25%_) fell marginally into the innovative area on the RPM map and could be considered an innovative sustainable plant−based dairy alternative given its considerably higher protein content, even though it needs some optimization to improve its protein content and amino acid profile. The color of the final formulation was important when defining sustainable innovative plant−based dairy alternatives as this impacted on the appearance liking of the product. Combining three plant raw materials was a useful approach to develop a sustainable plant−based dairy alternative that has high protein as well as positive sensory appeal. An optimum formulation may be developed based on any of these prototypes to enhance the nutritional content as well as the sensory appeal using the methods described in this study. From the current study, however, Prototype N (C_50%_, P_25%_, M_25%_) can be considered as the most innovative sustainable plant−based dairy alternative within this prototype set. A multi−blend approach to developing plant−based dairy alternatives may be more sustainable as this could reduce the over−reliance on a single plant raw material to provide sustainable foods with a good source of proteins. This also provide a natural means of improving the sensory appeal of such products without the need for adding flavours and with the added benefit of a better nutrient profile.

## Figures and Tables

**Figure 1 foods-10-00482-f001:**
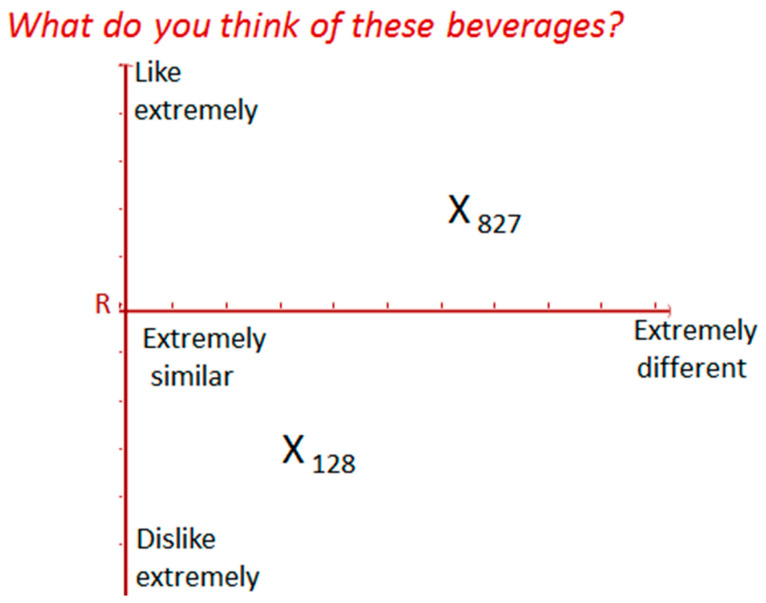
T-Map Scale used for RPM. X_128_ represents a mark (X) placed on the map by an assessor to show the position of product 128 on the T-Map Scale. This means that product 128 is perceived as liked less compared to the reference (R) and quite similar to the reference. Product 827 is perceived as liked more compared to the reference (R) and more different from the reference (R).

**Figure 2 foods-10-00482-f002:**
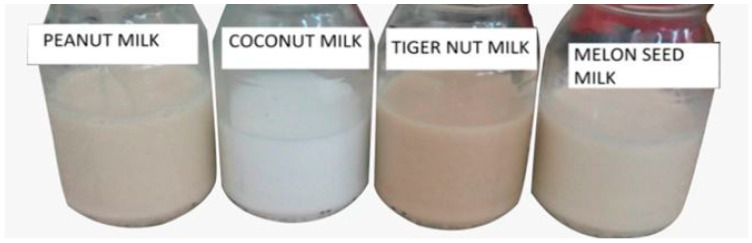
The different plant-based dairy alternatives: peanut milk, coconut milk, tiger nut milk and melon seed milk.

**Figure 3 foods-10-00482-f003:**
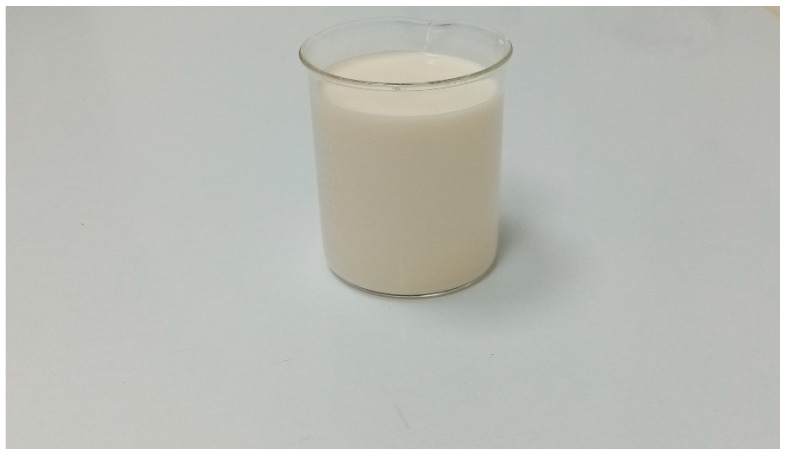
Even ultra-high temperature pasteurized (UHT) full cream milk used as reference sample, R and also as blind control (BC) for RPM test.

**Figure 4 foods-10-00482-f004:**
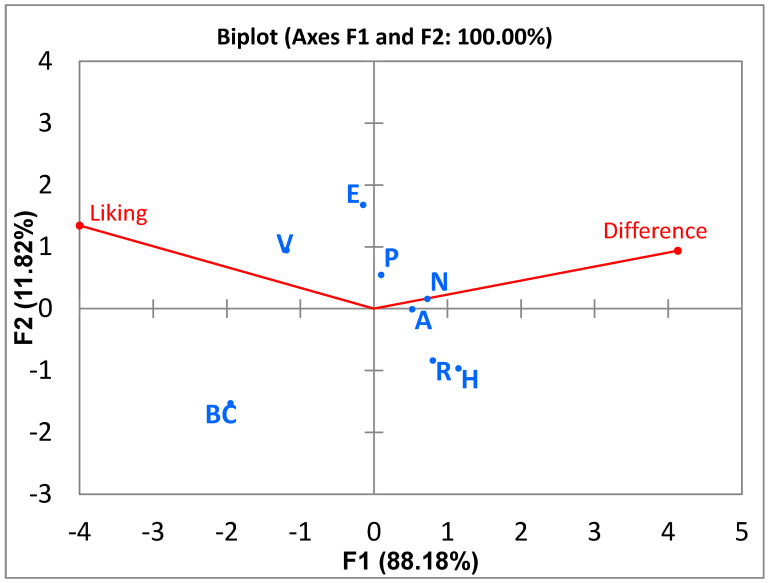
Product Map showing the area of innovation between the difference and liking axes. Legend: H—37.5% melon seed milk, 25% coconut milk, 37.5% peanut milk; A—25% melon seed milk, 50% coconut milk, 25% tiger nuts milk; P—50% coconut milk, 25% tiger nuts milk, 25% peanut milk; N—25% melon seed milk, 50% coconut milk, 25% peanut milk; R—37.5% melon seed milk, 25% coconut milk, 37.5% tiger nuts milk; E—25% coconut milk, 37.5% tiger nuts milk, 37.5% peanut milk; V*—Commercial plant-based dairy alternative and BC*— Even ultra-high temperature pasteurized (UHT) full cream milk; *—Commercial products.

**Figure 5 foods-10-00482-f005:**
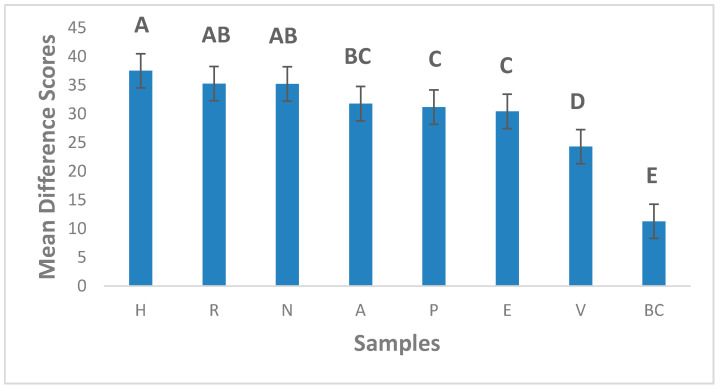
Mean difference from reference scores. Error bars are standard errors of the mean. Letters represent Fishers Fisher’s Least Significant Difference (LSD) Post hoc analysis of the means. Samples with no letters in common are statistically different at the 95% confidence level. Legend: H—37.5% melon seed milk, 25% coconut milk, 37.5% peanut milk; A—25% melon seed milk, 50% coconut milk, 25% tiger nuts milk; P—50% coconut milk, 25% tiger nuts milk, 25% peanut milk; N—25% melon seed milk, 50% coconut milk, 25% peanut milk; R—37.5% melon seed milk, 25% coconut milk, 37.5% tiger nuts milk; E—25% coconut milk, 37.5% tiger nuts milk, 37.5% peanut milk; V*—Commercial plant-based dairy alternative and BC*— Even ultra-high temperature pasteurized (UHT) full cream milk; *—Commercial products.

**Figure 6 foods-10-00482-f006:**
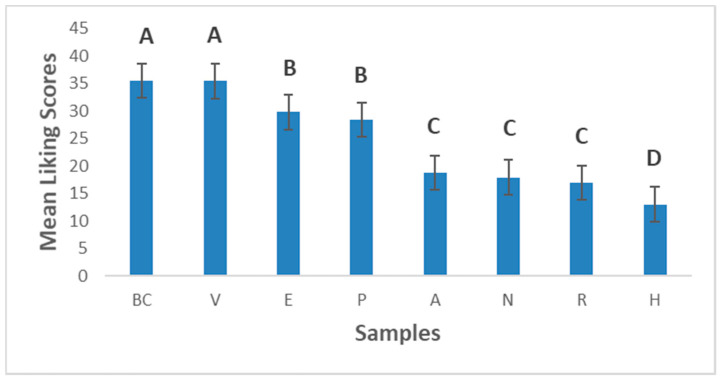
Mean liking compared to the reference. Error bars are standard errors of the mean. Letters represent Fishers Fisher’s Least Significant Difference (LSD) Post hoc analysis of the means. Samples with no letters in common are statistically different at the 95% confidence level. Legend: H—37.5% melon seed milk, 25% coconut milk, 37.5% peanut milk; A—25% melon seed milk, 50% coconut milk, 25% tiger nuts milk; P—50% coconut milk, 25% tiger nuts milk, 25% peanut milk; N—25% melon seed milk, 50% coconut milk, 25% peanut milk; R—37.5% melon seed milk, 25% coconut milk, 37.5% tiger nuts milk; E—25% coconut milk, 37.5% tiger nuts milk, 37.5% peanut milk; V*—Commercial plant-based dairy alternative and BC*— Even ultra-high temperature pasteurized (UHT) full cream milk; *—Commercial products.

**Figure 7 foods-10-00482-f007:**
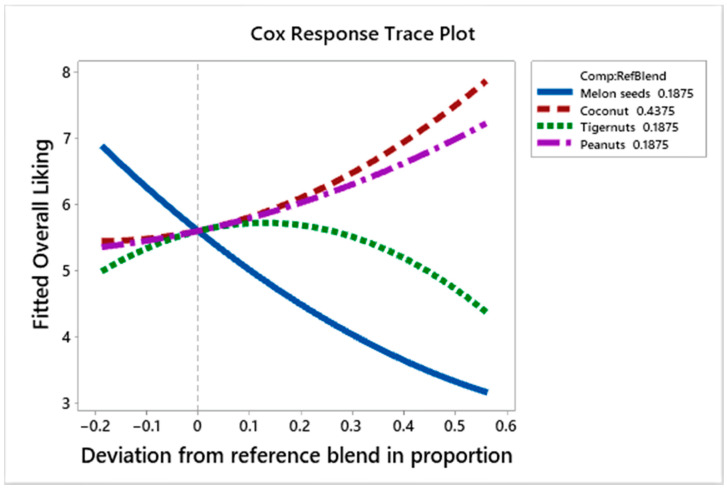
Cox response trace plot to show effect of ingredient on overall liking scores.

**Figure 8 foods-10-00482-f008:**
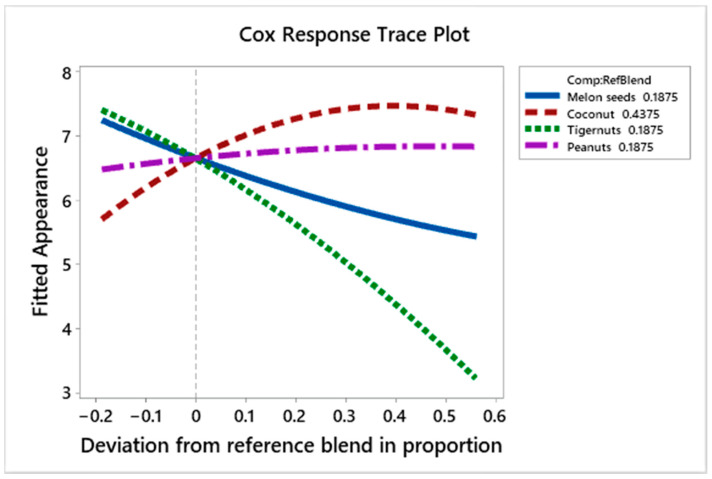
Cox response trace plot to show effect of ingredient on appearance liking scores.

**Figure 9 foods-10-00482-f009:**
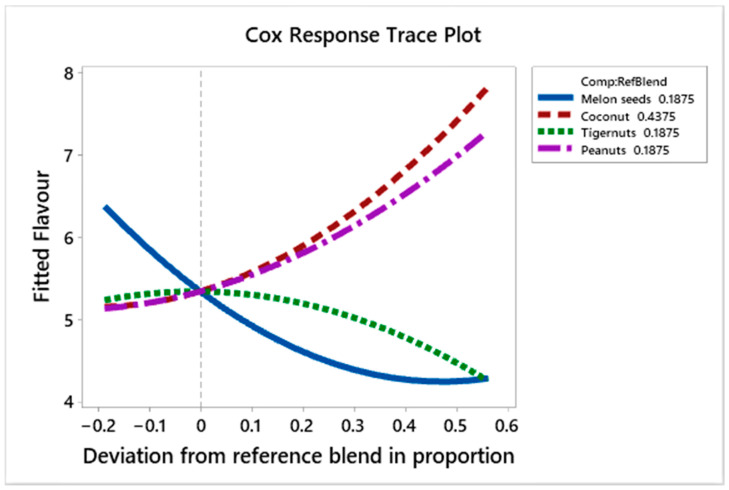
Cox response trace plot to show effect of ingredient on flavour liking scores.

**Figure 10 foods-10-00482-f010:**
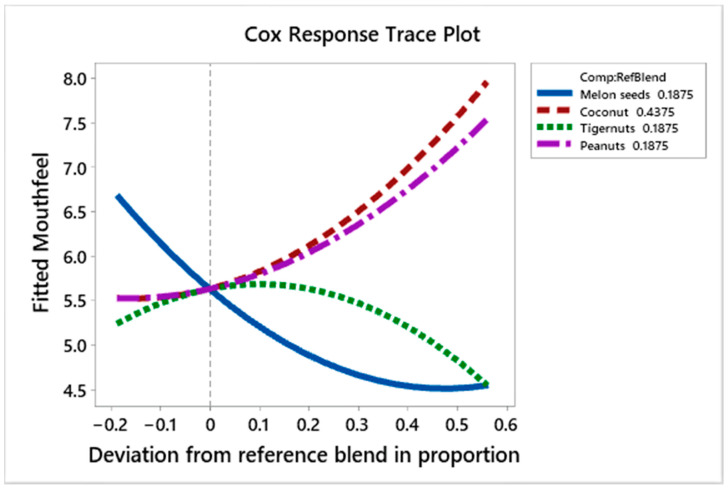
Cox response trace plot to show effect of ingredient on mouthfeel liking scores.

**Figure 11 foods-10-00482-f011:**
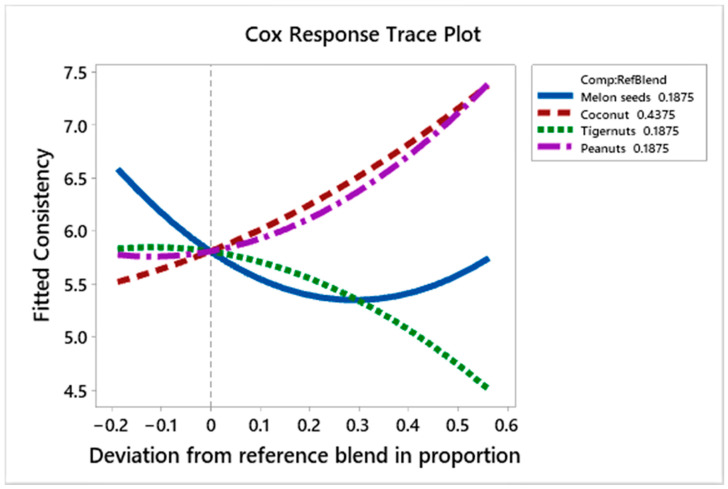
Cox response trace plot to show effect of ingredient on consistency liking scores.

**Figure 12 foods-10-00482-f012:**
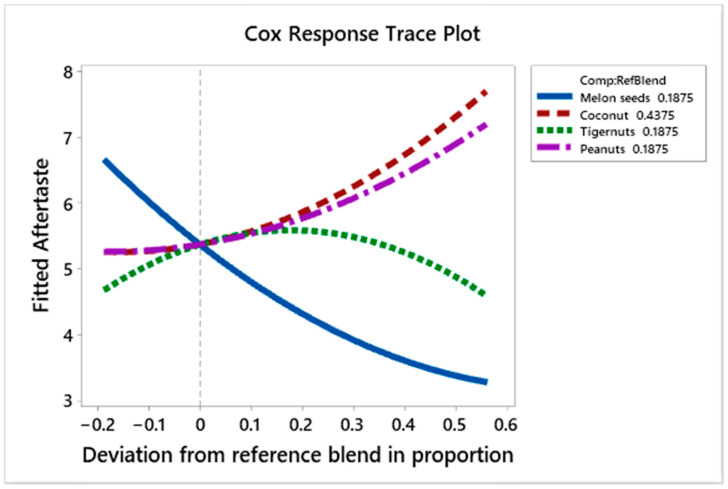
Cox response trace plot to show effect of ingredient on aftertaste liking scores.

**Table 1 foods-10-00482-t001:** Lower and upper limits of the four component raw materials.

Component	Lower Limit (%)	Upper Limit (%)
Coconut	25	100
Peanut	0	100
Tiger nut	0	100
Melon Seeds	0	100

**Table 2 foods-10-00482-t002:** Design matrix for a four-component mixture design for the plant-based dairy alternatives.

Run Order	Prototype	X_1_ (%)	X_2_ (%)	X_3_ (%)	X_4_ (%)	Total (%)
**1**	**A**	**50.000**	**0.000**	**25.000**	**25.000**	**100**
2	B	34.375	9.375	46.875	9.375	100
3	C	71.875	9.375	9.375	9.375	100
4	D	34.375	9.375	9.375	46.875	100
**5**	**E**	**25.000**	**37.500**	**37.500**	**0.000**	**100**
6	F	25.000	0.000	75.000	0.000	100
7	G	25.000	75.000	0.000	0.000	100
**8**	**H**	**25.000**	**37.500**	**0.000**	**37.500**	**100**
9	I	62.500	0.000	37.500	0.000	100
10	J	25.000	0.000	0.000	75.000	100
11	K	34.375	46.875	9.375	9.375	100
12	L	100.000	0.000	0.000	0.000	100
13	M	62.500	37.500	0.000	0.000	100
**14**	**N**	**50.000**	**25.000**	**0.000**	**25.000**	**100**
15	O	62.500	0.000	0.000	37.500	100
**16**	**P**	**50.000**	**25.000**	**25.000**	**0.000**	**100**
17	Q	43.750	18.750	18.750	18.750	100
**18**	**R**	**25.000**	**0.000**	**37.500**	**37.500**	**100**
19	S	25.000	25.000	25.000	25.000	100

X_1_—Coconut Milk, X_2_—Peanut milk, X_3—_Tiger nut milk, X_4_—Melon Seed Milk. Formulations in **bold font** are the 3-blend formulations used for consumer preference test using Relative Preference Mapping (RPM).

**Table 3 foods-10-00482-t003:** Products used for Relative Preference Mapping.

Product Number	Product Code	Product Details	Other Details
1	A	C _(50%)_ T _(25%)_ M _(25%)_	Prototype
2	E	C _(25%)_ P _(37.5%)_ T _(37.5%)_	Prototype
3	H	C _(25%)_ P _(37.5%)_ M _(37.5%)_	Prototype
4	N	C _(50%)_ P _(25%)_ M _(25%)_	Prototype
5	P	C _(50%)_ P_(25%)_ T_(25%)_	Prototype
6	R	C _(25%)_ T _(37.5%)_ M _(37.5%)_	Prototype
7	V	Vitamilk	Commercial product
8	BC	Even ultra-high temperature pasteurized (UHT) full cream milk	Commercial product sweetened to the same concentration as the prototypes

C—Coconut milk, T—Tiger nut milk, P—Peanut milk, M—Melon seed milk. Subscripts indicate the percentage component of the ingredient in the final formulation based on the mixture design method.

**Table 4 foods-10-00482-t004:** Proximate Composition of the products arranged in order of decreasing protein content.

Product	Composition (%)					
	Protein	Fat	Ash	Carbohydrates	Moisture	Total Solids
* BC	3.76 ± 0.00 ^a^	2.49 ± 0.22 ^f^	0.71 ± 0.02 ^a^	5.23 ± 0.06 ^e^	87.84 ± 0.01 ^b^	12.17 ± 0.02 ^e^
H	3.51 ± 0.02 ^ab^	6.30 ± 0.42 ^a^	0.25 ± 0.01 ^f^	4.68 ± 0.01 ^g^	85.26 ± 0.01 ^f^	14.74 ± 0.01 ^a^
* V	3.30 ± 0.04 ^b^	3.36 ± 0.00 ^e^	0.41 ± 0.01 ^cd^	6.04 ± 0.01 ^b^	86.89 ± 0.11 ^d^	13.18 ± 0.11 ^c^
R	2.31 ± 0.21 ^c^	4.09 ± 0.05 ^d^	0.51 ± 0.07 ^b^	5.90 ± 0.03 ^c^	87.19 ± 0.02 ^c^	12.81 ± 0.02 ^d^
N	2.16 ± 0.39 ^cd^	6.54 ± 0.04 ^a^	0.31 ± 0.02 ^ef^	5.09 ± 0.01 ^f^	85.89 ± 0.22 ^e^	14.23 ± 0.22 ^b^
A	1.89 ± 0.24 ^de^	5.11 ± 0.20 ^b^	0.49 ± 0.01 ^bc^	4.28 ± 0.01 ^h^	88.22 ± 0.13 ^a^	11.78 ± 0.13 ^f^
E	1.90 ± 0.09 ^de^	4.71 ± 0.08 ^c^	0.36 ± 0.02 ^de^	7.26 ± 0.01 ^a^	85.77 ± 0.01 ^e^	14.24 ± 0.01 ^b^
P	1.65 ± 0.06 ^e^	5.00 ± 0.35 ^bc^	0.34 ± 0.00 ^de^	5.65 ± 0.01 ^d^	87.36 ± 0.23 ^c^	12.64 ± 0.23 ^d^

Legend: H—37.5% melon seed milk, 25% coconut milk, 37.5% peanut milk; A—25% melon seed milk, 50% coconut milk, 25% tiger nuts milk; P—50% coconut milk, 25% tiger nuts milk, 25% peanut milk; N—25% melon seed milk, 50% coconut milk, 25% peanut milk; R—37.5% melon seed milk, 25% coconut milk, 37.5% tiger nuts milk; E—25% coconut milk, 37.5% tiger nuts milk, 37.5% peanut milk; V—Commercial plant-based dairy alternative and BC—Even ultra-high temperature pasteurized (UHT) full cream milk; * Commercial products. Values with the same letter (superscript) in a column are not statistically significantly different to each other.

**Table 5 foods-10-00482-t005:** Color indices of samples used in the RPM arranged in order of decreasing L * values.

Product	L *	A *	B *	ΔE
* BC	95.23 ± 0.10 ^a^	−0.93 ± 0.01 ^h^	13.66 ± 0.05 ^e^	12.01 ± 0.01 ^h^
N	85.54 ± 0.50 ^b^	0.52 ± 0.04 ^g^	12.74 ± 0.05 ^g^	16.64 ± 0.01 ^g^
* V	85.31 ± 0.02 ^b^	1.41 ± 0.01 ^b^	16.25 ± 0.04 ^a^	19.36 ± 0.01 ^e^
H	83.13 ± 0.32 ^c^	0.86 ± 0.02 ^e^	13.74 ± 0.06 ^d^	19.07 ± 0.06 ^f^
A	80.06 ± 0.77 ^d^	0.58 ± 0.02 ^f^	13.21 ± 0.05 ^f^	21.32 ± 0.01 ^d^
P	79.32 ± 0.05 ^e^	0.98 ± 0.05 ^c^	13.62 ± 002 ^e^	22.17 ± 0.01 ^c^
E	78.22 ± 0.13 ^f^	1.87 ± 0.03 ^a^	14.61 ± 0.01 ^c^	23.69 ± 0.01 ^b^
R	75.91 ± 0.13 ^g^	0.91 ± 0.01 ^d^	15.36 ± 0.04 ^b^	25.93. ± 0.01 ^a^

Legend: H—37.5% melon seed milk, 25% coconut milk, 37.5% peanut milk; A—25% melon seed milk, 50% coconut milk, 25% tiger nuts milk); P—50% coconut milk, 25% tiger nuts milk, 25% peanut milk; N—25% melon seed milk, 50% coconut milk, 25% peanut milk); R—37.5% melon seed milk, 25% coconut milk, 37.5% tiger nuts milk; E—25% coconut milk, 37.5% tiger nuts milk, 37.5% peanut milk; V *—Commercial plant-based dairy alternative) and BC *—Even ultra-high temperature pasteurized (UHT) full cream milk * Commercial products. Values with the same letter (superscript) within a column have no statistically significant difference from each other.

**Table 6 foods-10-00482-t006:** Regression coefficients for sensory scores.

Predictor Variable	Coefficients					
	Overall Liking	Appearance	Flavour	Mouthfeel	Consistency	Aftertaste
X1	7.874	7.332	7.824	7.967	7.382	7.713
X2	8.396	6.751	8.794	9.16	8.813	8.751
X3	2.354	1.137	3.1	2.98	3.353	2.865
X4	2.769	5.151	4.815	5.083	6.816	3.215
X1X2	−5.01	3.002	−6.302	−7.756	−4.589	−7.483
X1X3	4.669	6.251	0.754	2.135	3.131	3.241
X1X4	−8.923	4.815	−8.635	−8.272	−6.567 *	−7.945 *
X2X3	3.94	6.069	2.092	5.496	−0.187	6.739
X2X4	−1.749	−4.847	−4.927	−6.491	−6.724*	−6.028
X3X4	8.719	−5.55	1.339	2.61	−2.164	5.487
*R*^2^ (%)	79.91	89.33	82.81	88.44	90.55	93.88
*R*^2^ Adjusted (%)	59.82	78.67	65.63	76.88	81.11	87.76

(X1)—Coconut milk; (X2)—Peanut milk; (X3)—Tiger nuts milk and (X4)—Melon seed milk; * Significant (*p* ≤ 0.05).

**Table 7 foods-10-00482-t007:** Mean liking scores for the 19 formulations arranged in order of decreasing overall liking scores.

Product Code	Sensory Scores					
	Overall Liking	Appearance	Flavour	Mouthfeel	Consistency	Aftertaste
L (C_100%_)	7.89 ± 0.78 ^a^	7.11 ± 1.62 ^abcd^	7.78 ± 0.83 ^a^	7.78 ± 0.83 ^a^	7.33 ± 1.23 ^ab^	7.78 ± 0.97 ^a^
E (C_25%_, P_37.5%_, T_37.5%_)	7.44 ± 1.74 ^ab^	7.11 ± 1.45 ^abcd^	7.00 ± 1.80 ^ab^	7.44 ± 1.24 ^ab^	6.56 ± 2.46 ^abc^	7.00 ± 2.45 ^ab^
G (C_25%_, P_75%_)	7.44 ± 1.13 ^ab^	7.44 ± 1.33 ^abc^	7.22 ± 1.20 ^ab^	7.44 ± 1.88 ^ab^	7.44 ± 1.51 ^a^	7.11 ± 2.01 ^ab^
I (C_62.5%_, T_37.5%_)	7.22 ± 1.30 ^abc^	7.00 ± 1.23 ^abcd^	6.56 ± 1.81 ^abc^	6.89 ± 1.76 ^abc^	6.67 ± 1.66 ^abc^	6.67 ± 2.24 ^abc^
M (C_62.5%_, P_37.5%_)	7.00 ± 1.00 ^abcd^	8.11 ± 0.93 ^a^	7.00 ± 1.32 ^ab^	6.78 ± 1.79 ^abcd^	7.00 ± 2.00 ^abc^	6.44 ± 1.33 ^abc^
N (C_50%_, P_25%_, M_25%_)	6.55 ± 1.88 ^abcd^	7.56 ± 1.13 ^abc^	6.00 ± 2.00 ^abcde^	5.78 ± 2.91 ^abcde^	5.78 ± 2.77 ^abc^	5.11 ± 3.22 ^bcdef^
C (C_71.9%_, P_9.4%_, T_9.4%_, M_9.4%_)	6.55 ± 2.40 ^abcd^	7.78 ± 0.97 ^ab^	6.11 ± 2.42 ^abcd^	6.78 ± 2.49 ^abcd^	6.67 ± 2.65 ^abc^	6.11 ± 2.57 ^bcd^
A (C_50%_, T_25%_, M_25%_)	6.00 ± 0.50 ^bcde^	5.77 ± 1.30 ^def^	5.89 ± 0.83 ^abcde^	5.44 ± 2.13 ^bcde^	6.00 ± 2.35 ^abc^	5.78 ± 0.97 ^abcde^
B (C_34.4%_, P_9.4%_, T_46.9.%_, M_9.4%_)	5.78 ± 2.49 ^bcdef^	5.44 ± 1.24 ^ef^	4.78 ± 2.28 ^cdef^	5.56 ± 2.07 ^bcde^	5.67 ± 2.12 ^abc^	5.78 ± 2.86 ^abcde^
K (C_34.4%_, P_46.9%_, T_9.4%_, M_9.4%_)	5.68 ± 2.29 ^bcdef^	6.77 ± 1.72 ^abcde^	6.00 ± 2.55 ^abcde^	5.67 ± 2.40 ^bcde^	6.78 ± 1.79 ^abc^	6.00 ± 1.87 ^abcd^
Q (C_43.8%_, P_18.8%_, T_18.8%_, M_18.8%_)	5.44 ± 2.60 ^cdef^	6.22 ± 1.56 ^cde^	5.00 ± 2.74 ^cdef^	5.44 ± 2.88 ^bcde^	5.67 ± 2.65 ^abc^	5.22 ± 2.44 ^bcdef^
P (C_50%_, P_25%_, T_25%_)	5.33 ± 2.29 ^def^	6.44 ± 1.33 ^bcde^	5.33 ± 2.29 ^bcdef^	6.00 ± 1.73 ^abcde^	5.89 ± 1.76 ^abc^	6.00 ± 2.35 ^abcd^
S (C_25%_, P_25%_, T_25%_, M_25%_)	5.33 ± 2.12 ^def^	5.44 ± 2.00 ^ef^	5.00 ± 2.06 ^cdef^	5.33 ± 2.35 ^cde^	5.22 ± 2.05 ^bc^	5.56 ± 2.56 ^bcdef^
R (C_25%_, T_37.5%_, M_37.5%_)	4.44 ± 1.67 ^efg^	4.78 ± 1.64 ^fg^	4.44 ± 2.30 ^def^	5.11 ± 1.97 ^cde^	5.00 ± 2.60 ^c^	4.11 ± 1.69 ^defg^
F(C_25%_, T_75%_)	4.44 ± 2.83 ^efg^	3.67 ± 2.40 ^g^	4.33 ± 2.83 ^def^	4.44 ± 2.45 ^e^	4.88 ± 2.62 ^c^	4.67 ± 2.74 ^cdefg^
H (C_25%_, P_37.5%_, M_37.5%_)	4.11 ± 1.83 ^fgh^	6.44 ± 1.24 ^bcde^	5.00 ± 1.73 ^cdef^	5.11 ± 2.67 ^cde^	5.78 ± 2.82 ^abc^	3.78 ± 2.49 ^efg^
D (C_34.4%_, P_9.4%_, T_9.4%_, M_46.9%_)	4.00 ± 1.94 ^fgh^	6.67 ± 1.33 ^bcde^	3.78 ± 2.17 ^f^	4.11 ± 2.03 ^e^	5.33 ± 2.50 ^abc^	3.89 ± 1.83 ^efg^
O (C_62.5%_, M_37.5%_)	2.78 ± 2.10 ^gh^	7.67 ± 1.50 ^ab^	4.22 ± 2.49 ^def^	4.78 ± 3.03 ^de^	5.44 ± 2.46 ^abc^	3.67 ± 2.69 ^fg^
J (C_43.8%_, P_46.9%_, T_9.4%_, M_9.4%_)	2.56 ± 1.94 ^h^	6.44 ± 1.81 ^bcde^	4.11 ± 2.52 ^ef^	4.33 ± 2.64 ^e^	5.78 ± 2.77 ^abc^	3.00 ± 2.06 ^g^

Values with the same letters in a column (superscripts) do not show statistically significant difference.

## Data Availability

The data presented in this study are available on request from the corresponding author. The data are not publicly available due to commercialization potential of the data.

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
