# Peer review of "Sensory Acceptability and Proximate Composition of 3-Blend Plant-Based Dairy Alternatives"

_foods, 2021, doi:10.3390/foods10030482_

Round 1
Reviewer 1 Report
The introduction implies that dairy products have an adverse health effect due to their content of cholesterol or saturated fat, however, there is no current evidence supporting this view. You may want to rephrase this part that the plant-based dairy alternatives are preferred by some consumers due to their fat composition including their low level of saturated fat and the lack of cholesterol.
The authors list the common names for dairy alternatives such as “plant milk”, “plant milk alternatives”, however, it is important to mention that in some jurisdictions, the term “milk” cannot be used for non-dairy products (e.g., Canadian Food Inspection Agency specifies that milk is a product of lacteal secretion. Similarly, in EU, the term “milk” can be used for marketing and advertising of the products derived from animal milk. A similar legislation may be implemented soon in the United States after the review that is being conducted by U.S. Food & Drugs Administration.
Methods: Lines 151-152: “A total of 90 consumers of plant-based dairy alternatives participated in this study.” Please explain how the participants for this study were selected. Did they complete 24h food recall or Food Frequency Questionnaire, or simply indicated their preference towards plant-based dairy alternatives?
The major problem with the methodology is that the reference product (Even UHT Full Cream Milk) was altered (sweetened) and therefore the hedonic perception of this product may not reflect the natural taste and other sensory characteristics of milk. This should be indicated as a limitation. It is also unclear how the reference milk was sweetened? The following is stated (lines 156-157): “This product was sweetened to the same level as the prototypes to reduce bias caused by sweetness as a source of consumer appeal.” How was it done? Milk already contain lactose that provides a slight sweet taste. If the extra sugar was added in the same amount as to the other prototypes, this would create a stronger sweet taste compared to the other prototypes. On the other hand, the soy-based commercial dairy alternative product (“Vitamilk”) was used as is, because it is already sweetened, although it is not clear whether the sugar content and composition are the same as in the developed prototypes.
While the Results section is focused on the quantitative description of protein content in the developed samples, please comment on the amino acid composition (can be based on the literature) of the developed prototypes. It is known that the quality of milk protein is high due to the presence of all essential amino acids. As for the ash content, it would also be important to comment on the calcium content of all prototypes provided such information exists.
In Figure 2, please add the photograph of milk as a reference food.
Please add the table with 9-point hedonic scale scores (mean ± SD) for taste or pleasantness for all prototypes, soy beverage and milk.
Author Response
Thank you for the thorough review of our paper. We have addressed each of the comments raised and this can be found attached. We have highlighted the changes in the manuscript using tracked changes.
RESPONSES TO REVIEWER 1
Reviewer 1
Reviewer Comment-The introduction implies that dairy products have an adverse health effect due to their content of cholesterol or saturated fat, however, there is no current evidence supporting this view. You may want to rephrase this part that the plant-based dairy alternatives are preferred by some consumers due to their fat composition including their low level of saturated fat and the lack of cholesterol.
Response-Line 53-the statement that ‘consumers prefer plant-based dairy alternatives because they want to avoid cholesterol’ has been replaced with ‘who have a preference the vegan diet’. The line that says plant-based dairy alternatives are cholesterol free and contain mainly polyunsaturated fatty acids in lines 58-59 has been removed.
Reviewer Comment-The authors list the common names for dairy alternatives such as “plant milk”, “plant milk alternatives”, however, it is important to mention that in some jurisdictions, the term “milk” cannot be used for non-dairy products (e.g., Canadian Food Inspection Agency specifies that milk is a product of lacteal secretion. Similarly, in EU, the term “milk” can be used for marketing and advertising of the products derived from animal milk. A similar legislation may be implemented soon in the United States after the review that is being conducted by U.S. Food & Drugs Administration.
Response-It has been indicated from lines 47-50 that in the EU and Canada, the term milk can only be used for products from animals and the authors stated that they will use the appropriate term ‘plant-based dairy alternatives in the article’
Reviewer Comment-Methods: Lines 151-152: “A total of 90 consumers of plant-based dairy alternatives participated in this study.” Please explain how the participants for this study were selected. Did they complete 24h food recall or Food Frequency Questionnaire, or simply indicated their preference towards plant-based dairy alternatives?
Response-In lines 173-177, the authors have outlined the selection method used to recruit the participants for the consumer test using RPM. The participants who stated that they consumed a plant-based dairy alternative at least once a month were selected for the test
Reviewer Comment-The major problem with the methodology is that the reference product (Even UHT Full Cream Milk) was altered (sweetened) and therefore the hedonic perception of this product may not reflect the natural taste and other sensory characteristics of milk. This should be indicated as a limitation. It is also unclear how the reference milk was sweetened? The following is stated (lines 156-157): “This product was sweetened to the same level as the prototypes to reduce bias caused by sweetness as a source of consumer appeal.” How was it done? Milk already contain lactose that provides a slight sweet taste. If the extra sugar was added in the same amount as to the other prototypes, this would create a stronger sweet taste compared to the other prototypes. On the other hand, the soy-based commercial dairy alternative product (“Vitamilk”) was used as is, because it is already sweetened, although it is not clear whether the sugar content and composition are the same as in the developed prototypes.
Response- In lines 182-184, the procedure for sweetening the dairy milk sample has been outlined. In lines 449-451, it has been indicated that sweetening the dairy milk sample is a limitation as the dairy milk sample could taste sweeter than the prototypes.
Reviewer Comment-While the Results section is focused on the quantitative description of protein content in the developed samples, please comment on the amino acid composition (can be based on the literature) of the developed prototypes. It is known that the quality of milk protein is high due to the presence of all essential amino acids. As for the ash content, it would also be important to comment on the calcium content of all prototypes provided such information exists.
Response-The amino acid profiles of the various plant materials (based on literature) used for the prototypes has been discussed in lines 310-324. This was used to give an idea of the amino acid profiles of the prototypes.
The calcium contents of the various plant materials (based on literature) used for the prototypes has been discussed in lines 329-333. This showed that the prototypes will have low calcium contents as reported for most plant-based dairy alternatives in literature.
Reviewer Comment In Figure 2, please add the photograph of milk as a reference food.
Response- The picture of the reference sample (Even UHT Full Cream milk has been added in line 360
Please add the table with 9-point hedonic scale scores (mean ± SD) for taste or pleasantness for all prototypes, soy beverage and milk.
Response-Milk and soy milk were not included in the consumer test using the 9-point hedonic
Reviewer 2 Report
97. Incorporate more information about the focus group. How many people participated? Where was it held, how long was the discussion etc.
109. Reformulation requires a bullet point to be inconsistent with the other 2 points.
128. What is the reasoning for modifications from the existing method?
151. As per the guidelines of SSP, it is recommended to have at least 100 people for a sensory study. Do a power analysis to show if 90 consumers is enough.
Conclusion should be shortened and focus only on the end conclusions. No need to repeat the results here.
Author Response
Thank you for the thorough review. We have addressed all comments and also included the power analysis requested. This has added quality to our paper.
RESPONSES TO REVIEWER 2
Reviewer 2
Reviewer comment- 97. Incorporate more information about the focus group. How many people participated? Where was it held, how long was the discussion etc.
Response-Lines 102 -107- The organization of the focus group has been elaborated. The number of focus groups (3), the number of participants (17), the duration of each focus group (90mins) and the location (Sensory Evaluation Laboratory of the University of Ghana) has been indicated
Reviewer comment 109. Reformulation requires a bullet point to be inconsistent with the other 2 points.
Response-Line 118- Reformulation has been bulleted to be consistent with the first 2 points.
Reviewer comment –128. What is the reasoning for modifications from the existing method?
Response-Lines 142-148-The reasons for the modifications in the processing of melon seed milk and peanuts milk have been explained
Reviewer comment -151. As per the guidelines of SSP, it is recommended to have at least 100 people for a sensory study. Do a power analysis to show if 90 consumers is enough.
Response-Using the Exact method in SAS, with an alpha of 0.05, using the group means for the 8 samples and the standard deviation for the liking scores. The calculated power was 0.999 out of 1. Please see power analysis below:
|
The SAS System |
The POWER Procedure
Overall F Test for One-Way ANOVA
|
Fixed Scenario Elements |
|
|
Method |
Exact |
|
Alpha |
0.05 |
|
Group Means |
13.009 18.769 28.4 17.927 16.937 29.791 35.451 35.466 |
|
Standard Deviation |
16.3 |
|
Sample Size per Group |
90 |
|
Computed Power |
|
Power |
|
>.999 |
Reviewer comment-Conclusion should be shortened and focus only on the end conclusions. No need to repeat the results here.
Response-Lines 609-625- The conclusion has been shortened to state only the end conclusions, the results have been removed.